# UNCERTAINTY QUANTIFICATION WITH GENERATIVE-SEMANTIC ENTROPY ESTIMATION FOR LARGE LANGUAGE MODELS

## ABSTRACT

We introduce **Generative-Semantic Entropy Estimation** (GSEE), a model-agnostic algorithm that efficiently estimates the generative uncertainty associated with foundation models, while requiring no additional auxiliary model inference steps. In principle, for any foundation model input data, GSEE numerically estimates the uncertainty encapsulated in the internal, semantic manifold of the LLM generated responses to the input data. In this way, high uncertainty is indicative of hallucinations and low generative confidence. Through experiments, we demonstrate the superior performance of GSEE for uncertainty quantification (UQ) amongst state-of-the-art methods across a variety of models, datasets and problem settings, including: unbounded language prompting, constrained language prompting, variable generative stochasticity, acute semantic diversity prompting and as a barometer for hallucinations and predictive accuracy.

## 1 INTRODUCTION

In recent years, powerful foundation models, including Large Language Models (LLMs) (Naveed et al., 2023; Minaee et al., 2024) and Large Multi-Modal Models (LMMs) (Cui et al., 2024; Liu et al., 2023a) have ushered in a new epoch of multi-faceted, intelligent conversational agents. Despite their significant early successes and widespread use, foundation models nevertheless currently suffer from several critical shortcomings, including their lack of transparency and predilection for "hallucinations." (Huang et al., 2023) These deficiencies severely limit the trustworthiness and potential deployment of LLMs for real-world applications, particular in safety-critical domains. The widespread adoption and future success of LLMs and related models is critically dependent upon efforts to improve their transparency and explainability.

While the problem of uncertainty quantification (UQ) has enjoyed a rich and extensive research focus across many classical machine learning and deep learning (Gawlikowski et al., 2023) applications, LLM and related natural language generation (NLG) problem settings pose unique challenges to UQ. This difficulty primarily stems from: (i) the unbounded character of NLG and (ii) complexities surrounding efforts to effectively and reliably quantify semantic structure (Kuhn et al., 2023). In the former case, reliable uncertainty quantification for variable-length NLG tasks is complicated by a variety of issues, including ambiguity of model confidence probing, e.g., which output token(s) are most representative of model confidence, (Chen et al., 2024) and sensitivity to generative text length (Bengio et al., 2003) – to name but two critical issues. In addition, precise methods to capture model uncertainty are also complicated by the problem of how to ascertain *semantic self-consistency* for NLG (Malinin & Gales, 2021).

With these challenges in mind, we introduce **Generative-Semantic Entropy Estimation** (GSEE) a lightweight, model-agnostic algorithm to estimate UQ for LLM and other generative models. Our method renders predictive uncertainty in a mathematically-principled fashion, by numerically estimating the uncertainty encapsulated in the internal, semantic manifold of the LLM generated responses to the input data. GSEE follows a series of straightforward steps: (1) Given an input datum (text/text + image), we render multiple text outputs with the generative model through a single forward pass; (2) we then extract latent embeddings for each of these generated outputs; (3) next, we calculate the covariance matrix with respect to these mean-centered semantic embeddings; (4) lastly,

we define the model predictive uncertainty as the *spectral-entropy* of this covariance matrix, *viz.*, the entropy of the distribution of eigenvlues of the principal components of the generated outputs of the model. This measure is known to approximate the *effective dimension* of the semantic manifold spanned by the generated outputs (Zhang et al., 2024). Hence, here, larger spectral-entropy corresponds with larger semantic diversity in the generated outputs, and thus higher model uncertainty.

The current work provides the following contributions:

- We define a novel, mathematically-principled algorithm, GSEE, to approximate uncertainty quantification for LLM and related foundation models. GSEE operates on the internal states of an LLM, yielding a holistic measure of semantic self-consistency in model responses. Concretely, we leverage spectral-entropy to estimate the information content in the semantic manifold of generated responses.

- We test our algorithm against established UQ methods on a variety of essential experimental settings, including across LLM and LMM model types.

- We introduce several nuanced experimental conditions to benefit UQ understanding and hallucination detection for LLMs, including what we term: *constrained language prompting* and *high stochaticity conditioning*.

At a high level, improving uncertainty quantification for LLMs and related models can greatly expand the usefulness and applicability of these nascent models Zhao et al. (2024); Liu et al. (2023b). In particular, reliable UQ can help facilitate better LLM performance, enhance LLM-related human interaction and help foster trust in foundational model dependent systems. In terms of core algorithm advantages, GSEE does not require additional natural language inference (NLI) (Welleck et al., 2019) or related auxiliary model processing steps – boosting its relevance for real-time and computationally-constrained environments; GSEE is moreover model-agnostic and outperforms other baseline techniques on many challenging UQ tasks, as we show in our experimental results.

## 2 BACKGROUND AND PRIOR WORK

Much of the prior art for uncertainty quantification and hallucination detection with LLMs and related foundation models relies on the thesis that when a model is uncertain about its generated output, this answer distribution tends to exhibit high variability. In the domain of language generation specifically, this variation can be codified with respect to the notion of semantic consistency (Raj et al., 2023), predictive uncertainty, or entropy. As such, for example, a semantically differentiated answer distribution is more likely to be erroneous (or hallucinatory) than a semantically consistent distribution.

UQ methods for LLMs generally fall into two broad categories of black-box and white-box techniques for measuring semantic consistency, predictive uncertainty and variability in the generated answer distribution. Black-box techniques rely on analyzing these quantities with respect to the model outputs – this is to say, in the textual domain. Many such techniques exist (Lin et al., 2022; Wang et al., 2022), including, notably, *Lexical Similarity* (Liu et al., 2023a) which leverages a similarity measure using a bespoke natural language inference (NLI) model (*e.g.*, BERT (Devlin et al., 2018)) to assess semantic consistency across generated outputs by averaging similarity over generated output pairs. In this vein, much previous related work leverages NLI models to compare the semantic similarity of LLM-generated responses or some other NLI-related measure, including entailment, equivalence and consistency (Bowman et al., 2015). Two basic drawbacks of these prior methods are that they (i) require additional model compute/inference to approximate semantic similarity and (ii) the semantic similarity measure relies on extrinsic uncertainty estimates, independent of the LLM itself, thus adding undesirable noise to the UQ estimation process.

White-box models by contrast directly leverage internal representations of the generative process (e.g., hidden layer activations) to estimate predictive uncertainty. *Perplexity* (Ren et al., 2023) defines predictive uncertainty as the joint probability of output tokens for a single generated output. While Perplexity is an effective and widely-used measure of semantic consistency in literature, it is nevertheless known to suffer from brittleness due to the instability of token-level likelihoods (Wang et al., 2024). To mitigate these deficiencies, other techniques rely on multiple output NLG, where semantic diversity and related metrics account for larger sample outputs. The works closest to the

current work include (Chen et al., 2024) and (Wei et al., 2024). The former differs from the current work in two fundamental ways: (1) we include length normalization directly into our UQ metric in order to reduce semantic consistency sensitivity to output length; (2) because GSEE is calculated from spectral-entropy, our measure of uncertainty accounts for the (comparatively richer) holistic distribution of the semantic manifold and not simply the aggregation of eigenvalues over the NLG samples. While Wei et al. (2024) also calculate entropy, their metric is used with respect to the model latent space (whereas ours reflects entropy in generated model outputs); moreover they lever-age this UQ metric to assess the information content in training datasets for LLMs, and not UQ for NLG prediction.

## 3 GENERATIVE-SEMANTIC ENTROPY ESTIMATION

Formally, suppose we wish to approximate the predictive uncertainty (for hallucination detection, improved generative performance, *etc*.) of a large language – or multi-modal – foundational model, denoted $f(\cdot)$ with respect to an input language (or multi-modal) datum $\mathbf{x}$. To estimate predictive UQ, we generate multiple language outputs for the input data and then approximate the semantic diversity of the low-dimensional manifold encapsulating these generated outputs. A large semantic diversity indicates high model uncertainty. GSEE proceeds in four total steps:

(1) Generate a set of $M$ language responses for the input data (modulated *via* the temperature parameter of the model):

$$f(\mathbf{x}) \rightarrow \{\hat{\mathbf{y}}^{(i)}\}_{i=1}^{i=M} \tag{1}$$

(2) Next, we extract latent embeddings, denoted $\mathbf{z} \in \mathbb{R}^l$, where $l$ denotes the latent embedding dimension, from $f$ for each of these generated outputs:

$$\{\hat{\mathbf{y}}^{(i)}\}_{i=1}^{i=M} \rightarrow \{\mathbf{z}^{(i)}\}_{i=1}^{i=M} \tag{2}$$

GSEE is agnostic to the *locus* of latent embedding extraction for $f$. However, following current best practices, we extract rich semantic embeddings by averaging penultimate layer representations of generated output tokens.

(3) Next, we center these embeddings by mean-subtracting; we then calculate the covariance of these centered embeddings, where $\mathbf{Z} \in \mathbb{R}^{M \times l}$ represents the corresponding centered data matrix:

$$\text{Cov}\left(\{\mathbf{z}^{(i)}\}_{i=1}^{i=M}\right) = \mathbf{Z}^T \mathbf{Z} \tag{3}$$

We denote the covariance matrix above as $\Sigma \in \mathbb{R}^{M \times M}$.

(4) Finally, we define the uncertainty quantification of $f$ with respect to the input $\mathbf{x}$ as the **length-normalized spectral-entropy of** $\Sigma$, which we define as the entropy of the normalized spectrum of $\Sigma$, denoted $\mathcal{H}(\Sigma_N)$:

$$UQ(f(\mathbf{x})) = \mathcal{H}(\Sigma_N)/\mu_{\mathbf{y}} \tag{4}$$

Above, $\mu_{\mathbf{y}}$ is defined as the mean length of the generated output set $\{\hat{\mathbf{y}}^{(i)}\}_{i=1}^{i=M}$. We include length-normalization in 4 to reduce our UQ formulation sensitivity to textual length (Raj et al., 2023).

To gain a better intuition as to why the calculation above helps quantify model uncertainty, recall that resolving the eigendecomposition of a centered covariance matrix is equivalent to solving the formal principal components analysis (PCA) problem for dimensionality reduction (Abdi & Williams, 2010). Thus in 4, we compute the entropy of the spectrum of $\Sigma$, which is to say we are approximating the "information content", *viz*., the effective subspace rank of the semantic manifold (Roy & Vetterli, 2007) encapsulated by the set: $\{\mathbf{z}^{(i)}\}_{i=1}^{i=M}\}$. Simply put, the more "semantically diverse" the set of outputs for the generative model, the higher the predictive uncertainty.

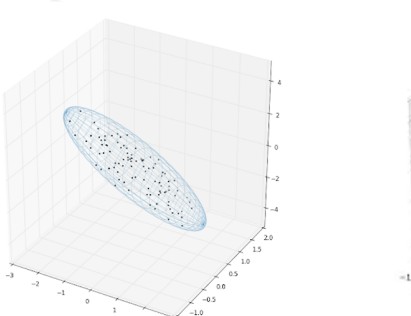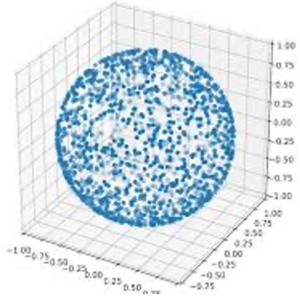

Figure 1: Conceptualization of non-Isotropic/Isotropic Semantic Manifolds for UQ. (Left) Non-Isotropic semantic manifold indicating differentiated principal components, yielding low spectral-entropy; (Right) Isotropic semantic manifold exhibiting large spectral entropy.

It is helpful to visualize an idealization of the preceding ideas; we provide this visualization in Figure 1; experimental evidence supporting this postulate is given in Figure 2. When an LLM is more certain of its response to input data, it tends to produce a set of semantically unified responses (this result is well-known in research literature (Lin et al., 2022)). Conversely, much like their human counterparts, LLMs tend to "flub" when they are uncertain. Thus, when a generative model is confident in its reply, a small number of differentiated, principal axes emerge in the semantic feature space, giving rise to a non-isotropic semantic manifold (yielding small spectral-entropy), whereas the absence of major semantic axes tends to give rise to isotropic manifolds exhibiting large spectral-entropy.

## 4 EXPERIMENTAL RESULTS

### 4.1 EXPERIMENTAL SETUP

To validate the effectiveness of GSEE, we ran a variey of UQ-related experiments, including: conventional experiments aimed to evaluate UQ performance against baseline LLM UQ methods (Section 4.2), novel, supplemental UQ experiments (Section 4.3) and ablation studies (Section 4.4).

**Models.** Our experiments comprise four different open-source LLM and LMM model types, including: GPT2-xl (Radford et al., 2019) and GPT3 (Brown et al., 2020) LLMs, and TinyLLaVA-3.1B (Jia et al., 2024) (with Phi-2 LLM backbone), TinyLLaVA-2.0B (with Stable-LM2 LLM backbone) LMMs. In each case, we employ pre-trained Hugging Face instances of these models, with no fine-tuning applied.

**Datasets.** We use four standard datasets for our experimental evaluations: Stanford Question Answering Dataset (SQuAD) (Rajpurkar et al., 2016), a reading comprehension dataset consisting of question-answer pairs culled from Wikipedia articles, Conversational Question Answering Dataset (CoQA) (Reddy et al., 2019), consisting of question-answer pairs generated from conversations, as well as the MM-Vet benchmark (Yu et al., 2023) and LAVA-Bench (in-the-wild) datasets (Li et al., 2024) which evaluate LMMs in terms of their integrated vision-language capabilities.

**Evaluation Metrics.** We use standard evaluation metrics from uncertainty estimation literature (Lin, 2004). In particular, we calculate the semantic agreement between the NLG output of the aforementioned LLM and LMMs with the ground-truth answer from the preceding dataset question-answer and question+image-answer pairs as a measure of correct NLG prediction. Concretely, we use both Rouge-L (Lin, 2004) and Semantic Similarity (SS) (Reimers & Gurevych, 2019) metrics to quantify agreement between NLG prediction ($\hat{\mathbf{y}}$) and ground-truth answers ($\mathbf{y}$), *i.e.,* Rouge-L($\mathbf{y}, \hat{\mathbf{y}}$) and SS($\mathbf{y}, \hat{\mathbf{y}}$). Next, we report the absolute magnitude of the Pearson Correlation Coefficient (PCC) between these Rouge-L and Semantic Similarity scores and the model uncertainty score, respectively. A large magnitude PCC value indicates strong correlation between the UQ method and the (in-)correctness of the NLG prediction; thus, large correlation implies that the UQ method is a good predictor of hallucinations and generative accuracy.

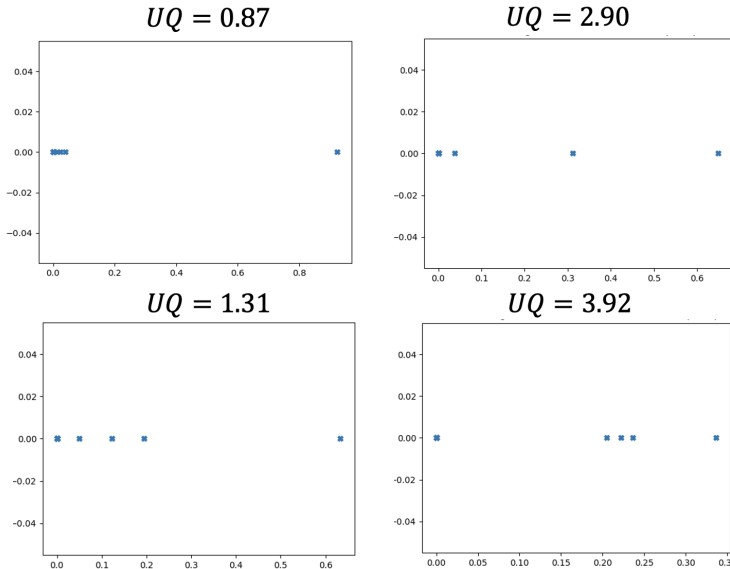

Figure 2: Visualizations of normalized eigenvalue distributions and corresponding GSEE-based UQ scores, using TinyLLaVA-3.1B with test examples culled from the MM-Vet dataset, where $M = 5$. (Left column) represents "good" prompt results; (Right column) shows "bad" prompt results from constrained language prompting experiments. In the former case, we observe instances of stronger semantic self-consistency as evidenced by a non-isotropic morphology.

**Baseline Methods.** We compare GSEE with the several popular UQ techniques, including: Eigen-Score (Chen et al., 2024), Lexical Similarity , Perplexity and Length-Normalized Entropy (LN-Entropy). Of note, we evaluate Eigenscore without test time clipping, as this technique is more or less orthogonal to the core technique presented in (Chen et al., 2024), and thus can be applied to potentially enhance any UQ method, including GSEE.

**Supplemental and Ablation Experiments.** In addition to the principal evaluation metric disclosed above, we furthermore evaluate GSEE in more diverse NLG settings, including with constrained language prompting and high stochaticity conditioning, that we detail below. We also perform ablation studies with respect to the number of generated responses $(M)$ and generative model temperature.

**Implementation.** Experiments are executed using Pytorch and transformers libraries and open-source, pre-trained models and tokenizers provided by Hugging Face. We use the following parameter settings for our baseline experiments: top-p$= 0.99$, num-beams$= 1$, $M = 5$, temperature$= 0.5$ and max-new-tokens$= 512$.

## 4.2 MAIN RESULTS

In Table 1 we report results for LMM UQ comparison with baseline techniques on the MM-Vet dataset. In each case, GSEE outperformed the baseline methods.

Despite these promising results, it is well-known that Rouge-L and Perplexity frequently generate noisy outcomes, sometimes yielding unreliable measures of semantic similarity, particular in cases involving long and/or complex textual passages. As evidence, we found the PCC measure between Rouge-L and Semantic Similarity on the MM-Vet ground-truth answers themselves to be only $0.68$. To better understand the performance of GSEE for uncertainty quantification in a broader setting, we therefore devised a novel set of additional experiments that capture more lucid distinctions between hallucination rendering and correct/incorrect generative results. We designate these experiments "constrained language prompting" and "high stochasticity conditioning."

Table 1: Experimental results for PCC between UQ methods and predictive "correctness" using Semantic Similarity and Rouge-L similarity on the MM-Vet dataset.

| Dataset: MM-Vet | | | | |
|---|---|---|---|---|
| UQ Method | TinyLLaVA-3.1B | | TinyLLaVA-2.0B | |
| Perplexity | 0.09 | 0.09 | 0.07 | 0.02 |
| LN-Entropy | 0.29 | **0.24** | 0.11 | 0.02 |
| EigenScore | 0.04 | 0.17 | 0.04 | 0.16 |
| Lexical Similarity | 0.23 | 0.04 | 0.16 | 0.08 |
| **GSEE (ours)** | **0.35** | **0.24** | **0.30** | **0.18** |

## 4.3 CONSTRAINED LANGUAGE PROMPTING AND HIGH STOCHASTICITY CONDITIONING

For constrained language prompting, we alter the prompting strategy so that for each ground-truth prompt-answer (or prompt+image-answer) pair in a dataset, we augment the prompt set with both "good" and "bad" prompts. More precisely, we stipulate:

- **Good prompt**: ground-truth prompt from dataset
- **Bad prompt**: "(You should ignore the input image). Say something random and incoherent in 1-2 sentences."

The purpose of introducing this constrained language prompting paradigm is to more directly control against noisy hallucination detection and errors in generative prediction correctness that otherwise degrade UQ evaluation processes. We assert that using this dichotomized prompting strategy supports this aim, as it enables a clearer delineation between correct and incorrect NLG responses. Tables 2 and 3 summarize GSEE performance for constrained language prompting experiments. Overall, GSEE demonstrated best performance compared to baseline models across the test datasets, with particularly strong results on the MM-Vet and CoQA datasets. On the Lava-Bench (in-the-wild) dataset, however, GSEE results were somewhat degraded. We believe that this phenomenon can be attributed, in part, to the relatively small size of this dataset.

To further assess the ability of GSEE to differentiate correct vs. incorrect NLG, we record summary statistics of the UQ score across the preceding experiments. These results are shown in Table 4. Our experiments demonstrate the ability of GSEE to capture semantic self-consistency (and its absence) in a generalized setting, across all four of our diverse experimental datasets.

We also tested the effectiveness of GSEE in a large semantic diversity setting, which we term high stochasticity conditioning. Concretely, we use the following prompt strategy:

- **Good prompt**: "In 1-2 sentences: describe in fine detail what you see in the image." (for single input image)
- **Bad prompt**: "In 1-2 sentences: describe in fine detail what you see in the image." (for $M$ randomly sampled image from the dataset)

The results of these experiments are shown in Table 5; GSEE provides a strong signal for testing semantic self-consistency with high stochasticity conditioning. For "bounded" NLG length experiments, we constrain both good and bad prompts to 1-2 sentences, whereas with "unbounded" NLG length experiments, we do not include this stipulation. These experiments are intended to evaluate GSEE under conditions of acute semantic diversity and to furthermore test robustness to NLG length.

## 4.4 ABLATION EXPERIMENTS

We additionally performed ablation studies with respect to generative model temperature and $M$, the number of language samples produced by the NLG model from the input data, see Table 6. From these experiments, we note that as the temperature increases, the correlation between the GSEE-based uncertainty score and "correct" NLG response generally increases. However, for very high temperatures (*e.g.*, $t = 1.0$) this correlation improvement saturates. For ablation on $M$, we observe that GSEE performance generally scales favorably as $M$ increases, which we believe to be sensible,

Table 2: Experimental results for PCC between UQ methods and predictive "correctness" using Semantic Similarity and Rouge-L similarity with constrained language prompting for LMMs on the MM-Vet and Lava-Bench (in-the-wild) datasets.

| Dataset: MM-Vet | | | | |
|---|---|---|---|---|
| UQ Method | TinyLLaVA-3.1B | | TinyLLaVA-2.0B | |
| | SS | Rouge-L | SS | Rouge-L |
| Perplexity | 0.17 | 0.23 | 0.23 | 0.28 |
| LN-Entropy | 0.36 | 0.33 | 0.31 | 0.40 |
| EigenScore | 0.07 | 0.04 | 0.05 | 0.15 |
| Lexical Similarity | 0.20 | 0.24 | 0.20 | 0.26 |
| **GSEE (ours)** | **0.56** | **0.52** | **0.52** | **0.53** |
| Dataset: Lava-Bench (in-the-wild) | | | | |
| UQ Method | TinyLLaVA-3.1B | | TinyLLaVA-2.0B | |
| | SS | Rouge-L | SS | Rouge-L |
| Perplexity | 0.20 | 0.18 | 0.53 | 0.27 |
| LN-Entropy | **0.73** | 0.32 | **0.66** | 0.65 |
| EigenScore | 0.16 | 0.56 | 0.22 | 0.60 |
| Lexical Similarity | 0.54 | **0.84** | 0.60 | **0.82** |
| **GSEE (ours)** | 0.56 | 0.76 | 0.61 | 0.70 |

Table 3: Experimental results for PCC between UQ methods and predictive "correctness" using Semantic Similarity and Rouge-L similarity with constrained language prompting for LLMs on the SQuAD and CoQA datasets.

| Dataset: SQuAD | | | | |
|---|---|---|---|---|
| UQ Method | GPT2-xl | | GPT3 | |
| | SS | Rouge-L | SS | Rouge-L |
| Perplexity | 0.31 | 0.06 | **0.54** | **0.12** |
| LN-Entropy | 0.40 | 0.03 | 0.41 | 0.10 |
| EigenScore | 0.08 | **0.10** | 0.18 | 0.06 |
| Lexical Similarity | 0.52 | 0.09 | 0.53 | **0.12** |
| **GSEE (ours)** | **0.59** | 0.08 | **0.54** | **0.12** |
| Dataset: CoQA | | | | |
| UQ Method | GPT2-xl | | GPT3 | |
| | SS | Rouge-L | SS | Rouge-L |
| Perplexity | 0.22 | 0.07 | 0.27 | 0.05 |
| LN-Entropy | 0.53 | 0.01 | 0.62 | 0.08 |
| EigenScore | 0.37 | 0.05 | 0.26 | 0.02 |
| Lexical Similarity | 0.65 | 0.07 | 0.66 | 0.11 |
| **GSEE (ours)** | **0.75** | **0.09** | **0.73** | **0.12** |

as the detection of semantic self-consistency should naturally improve with larger generative samples (comparable gains were however less pronounced for baseline UQ methods). We understand this favorable relationship between GSEE UQ efficacy and generative sample size as an auspicious feature for GSEE. We plan to further explore and optimize the interplay between sample generation size and UQ quality for GSEE in future work.

## 5 CONCLUSION

We presented Generative-Semantic Entropy Estimation, a lightweight, model-agnostic algorithm to estimate UQ for LLM/LMM and other generative models through the spectral-entropy of generated outputs. Per our experiments, GSEE performed best overall against baseline UQ methods in a variety of essential and diverse settings, including unbounded language prompting, constrained language prompting, high/low generative stochasticity and as a barometer for hallucination/predictive accuracy. With the growing need to improve trust and explainability for foundation models, we believe that GSEE and related methods can help facilitate better LLM performance, enhance LLM-related

Table 4: Results for summary statistics, including the mean and standard deviations of constrained language prompting experiments across datasets.

| Dataset | UQ($\mu$) | | UQ($\sigma$) | |
|---|---|---|---|---|
| | Good Prompt | Bad Prompt | Good Prompt | Bad Prompt |
| MM-Vet | 0.51 | 2.05 | 0.47 | 0.50 |
| Lava-Bench (in-the-wild) | 0.88 | 1.09 | 0.21 | 0.14 |
| SQuAD | 0.14 | 0.22 | 0.05 | 0.02 |
| CoQA | 0.15 | 0.22 | 0.02 | 0.02 |

Table 5: High stochasticity conditioning experimental results; we perform two core sets of experiments: with and without bounded NLG length.

| Experiment: Bounded NLG Length | | | |
|---|---|---|---|
| Prompt Type | UQ($\mu$) | UQ($\sigma$) | Avg NLG length |
| Good Prompt | 1.75 | 0.65 | 134.6 |
| Bad Prompt | 3.82 | 0.23 | 138.4 |
| Experiment: Unbounded NLG Length | | | |
| Good Prompt | 2.50 | 0.62 | 945.6 |
| Bad Prompt | 3.79 | 0.29 | 944.1 |

human interaction and bolster confidence in real-world, foundational model dependent systems. As computational resources become more streamlined and plentiful for the operation of foundation models in the near future, we believe that such scalable XAI solutions will become a *sine qua non* for deployed AI systems.

Table 6: Ablation study results on MM-Vet using TinyLLaVA-3.1B. (Left) Ablation on generative temperature; (Right) ablation on the number of NLG outputs ($M$).

| Temperature | SS | Rouge-L |
|---|---|---|
| 0.1 | 0.08 | 0.24 |
| 0.3 | 0.22 | 0.32 |
| 0.5 | 0.24 | 0.35 |
| 1.0 | 0.24 | 0.32 |

| $M$ | SS | Rouge-L |
|---|---|---|
| 5 | 0.24 | 0.35 |
| 10 | 0.27 | 0.35 |
| 20 | 0.27 | 0.34 |
| 50 | 0.30 | 0.40 |

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
