# OpenReview forum: "Uncertainty Quantification with Generative-Semantic Entropy Estimation for Large Language Models"
_ICLR.cc/2025/Conference — Submitted to ICLR 2025_

### Official Review · Reviewer_KNZL · 2024-10-22

**Soundness:** 1
**Presentation:** 1
**Contribution:** 2
**Rating:** 1
**Confidence:** 4

**Summary:**

The paper introduces GSEE, a method to estimate the semantic uncertainty of (vision) language models.
The method uses information in the internal state of the model to measure semantic uncertainty.
It is evaluated on four benchmark tasks, two pure language, two vision language tasks, using four different model architectures.

**Strengths:**

The idea of using the entropy instead of the determinant of the covariance matrix is a reasonable extension of the prior work of Chen et al.
The authors investigate both pure language as well as vision language models.

**Weaknesses:**

### Major:

* The framing of GSEE being model agnostic is misleading, as it is complety reliant on the quality of the extracted answer embeddings.
I am actually not aware of any method for UQ in LLMs/LMMs that would not be model agnostic as by this definition.
* I don't see how step (1) for GSEE in the introduction (line 51) should ever by possible.
It states that "Given an input datum ... we render multiple text outputs with the generative model through a single forward pass;".
I am not aware of any method to generate even one output sequence (consisting of more than one token) with a single forward pass, let alone multiple.
How it is described in line 126, it seems the usual setting of generating M sequences is used, with the expense of M forward passes times the lengths of individual sequences.
* How is GSEE "mathematically-principled" as claimed in line 61? There is no derivation of any technical analysis, solely an empirical evaluation of an ad-hoc measure.
This is not necessarily bad, if it works well in practice, however the claim of being principled is definately not backed up by what is presented in the paper.
* Since when is GPT3 an open-source LLM (line 196/197), available through HF transformers?
* I am completely lost in understanding section 4.3.
First, I don't understand the basic rationale behind introducing those "bad prompt" settings.
Second, why is there a need to "bound" prompts?
Aren't they given a priori, thus are completely under control of the experimentor?
Third, why are those experiments evaluated using summary statistics over the proposed uncertainty measures and do not use the PCC as for the main experiments?
* Critical experimental details are missing. E.g. what output sequence (of the M sequences) is actually used to evaluate correctness? The most likely one?
* How does one know, that the latent representation captures semantics?
Is there any investigation regarding this, e.g. comparing GSEE scores for handcrafted lexically diverse but semantically similar vs. semantically diverse but lexically similar answers?
Otherwise, it is a very strong claim that the latent space captures the semantics of a sentence well and that the eigenvectors correspond to different semantics rather than lexical structure, which I don't buy without evidence.
* In the ablation section, line 365, there is a statement that "comparable gains were however less pronounced for baseline UQ methods".
I didn't find the corresponding results anywhere, table 6 ony provides results for GSEE.
* The choice of baselines is very limited. There is a lot of work on semantic entropy (Kuhn, Aichberger, Farquhar, Duan, Bakman) that has very good performance but is not compared to.
Looking at the paper of Chen et al., which is very heavily cited reveals that they used those same baselines without adapting to improvements in those last two years.
Also, length normalization for predictive entropy is not without debate, adding the non normalized variant costs basically nothing yet would give more credibility to the experiments.
* Many things are not defined, e.g. is $\Sigma_N$ the same as $\Sigma$ or is there any difference?
Is $\Sigma_N$ normalized to make it a "probability distribution" for calculating the entropy?
How is the entropy actually calculated, e.g. only upper triangle or full matrix flattened?
* I don't get the reason for dividing the entropy term by the mean length.
The semantic embeddings $z$ are already averaged, thus do not contain any length information?
This term rather induces length information in the uncertainty measure.

### Minor:

* Overall, the manuscript is very hard to follow.
Consider streamlining the choice of wording, I often had to stop thinking about whether or not two things actually mean the same or not (e.g. prompting and conditioning in section 4.3).
* The naming "Generative-Semantic entropy estimation" is not optimal, given the well established "Semantic Entropy" method (Kuhn, Aichberger, Farquhar).

### Remarks

* Please improve the visuals of figure 2
* Please take a look at the style guide for tables, there is no need for vertical lines
* line 55: eigenvlues -> eigenvalues

**Questions:**

See major weaknesses. Additionally:

* Most prior work (e.g. Kuhn, Aichberger, Farquhar, Bakman, Duan) except Chen et al., this work is heavily based upon, uses AUROC/AUPR of being correct as evaluation metric for their uncertainty measure.
Why is using the PCC better than those established measures?

---
## References:

Chen, C., Liu, K., Chen, Z., Gu, Y., Wu, Y., Tao, M., ... & Ye, J. (2024). INSIDE: LLMs' Internal States Retain the Power of Hallucination Detection. ICLR24

Kuhn, L., Gal, Y., & Farquhar, S. (2023). Semantic uncertainty: Linguistic invariances for uncertainty estimation in natural language generation. ICLR23

Aichberger, L., Schweighofer, K., Ielanskyi, M., & Hochreiter, S. (2024). Semantically Diverse Language Generation for Uncertainty Estimation in Language Models. arXiv

Farquhar, S., Kossen, J., Kuhn, L., & Gal, Y. (2024). Detecting hallucinations in large language models using semantic entropy. Nature

Bakman, Y. F., Yaldiz, D. N., Buyukates, B., Tao, C., Dimitriadis, D., & Avestimehr, S. (2024). MARS: Meaning-Aware Response Scoring for Uncertainty Estimation in Generative LLMs. arXiv

Duan, J., Cheng, H., Wang, S., Wang, C., Zavalny, A., Xu, R., ... & Xu, K. (2023). Shifting attention to relevance: Towards the uncertainty estimation of large language models. ACL

---

> ### Author Response · Authors · 2024-11-27
> **Authors reply to Reviewer KNZL**
>
> We thank the reviewer for their detailed and comprehensive feedback.  In reply:
>
> 1. Yes, this is a fair distinction. Our comment was meant to imply that the GSEE algorithm is, in point of fact, independent of the latent embedding locus (as any model embedding can be used in principle) — however, the quality of the UQ will be sensitive to the choice of model and locus, naturally.
> 2. Here we simply meant to indicate by a nominal “forward pass”, i.e., with respect to an <end> token, multiple generations can be rendered; we address this computational cost in reviewer comments above. Thank you for your comment, we agree that this idea should be stated more clearly.
> 3. By “mathematically-principled” we underscore that GSEE corresponds with the effective rank of the covariance matrix of the generated responses, a well-established statistical measure of uncertainty.
> 4. Fair point, however, there are extant, publicly-available replications of GPT-3 + openAI tokenizers used in the HF community.
> 5.  Because semantic similarity is inherently difficult to measure consistently, yielding misleading conclusions regarding UQ specifically, we provide prompts, “bad” ones, that constrain the problem domain into objectively similar/dissimilar semantics settings, which yields a qualitatively different measure of UQ “correctness”. Using PCC in the latter setting would reintroduce the reliance upon noisy semantic similarity measures, e.g., Rouge-L and perplexity, so we instead opted to demonstrate the quality of good/bad prompting UQ cluster separation.
> 6. Thank you for the question. We use the average of the generated responses to evaluate correctness; we will make this point clearer in the final version of the paper.
> 7. Thank you for this important comment. The ability of LLMs to capture latent semantics is, to the authors’ knowledge, well-established, please see: Tennenholtz et al., “Demystifying Embedding Spaces using Large Language Models”, ICLR 2024.
> 8. We did not include these results in the table, as we felt they were provided few insights, as indicated by the attendant comment in the paper. Nevertheless, we are happy to include them in the final version of the paper.
> 9. This is a reasonable point. We did not generate non-length normalized results, but this is certainly something that we could include in a supplemental section of the paper.
> 10. Yes, the eigenvalues are normalized to yield a proper probability distribution; the entropy is calculated with respect to the full matrix.
> 11. We divide the entropy to reduce length sensitivity, but the reviewer makes a cogent point that this step introduces length information; a viable alternative could be to apply GSEE to processed outputs (e.g., reduced verbosity) and to subsequently remove the averaging according to mean length.
> 12. We will make a concerted effort to improve the streaming of word choice, thank you for the suggestion.

---

### Official Review · Reviewer_4pJg · 2024-11-03

**Soundness:** 3
**Presentation:** 3
**Contribution:** 2
**Rating:** 5
**Confidence:** 4

**Summary:**

This paper introduces Generative-Semantic Entropy Estimation (GSEE) to efficiently estimate uncertainty in generative models without requiring auxiliary model inference steps. GSEE numerically captures the uncertainty based on the spectral entropy of the covariance matrix of the generated outputs, linking high uncertainty to potential hallucinations and low confidence. The authors demonstrate that GSEE outperforms other uncertainty quantification (UQ) methods across various prompting conditions and models

**Strengths:**

(1) The effectiveness of applying GSEE to measure the semantic self-consistency in model responses has been demonstrated by extensive experiments on LLMs and LMMs.

(2) It is novel to design the experiments “constrained language prompting” and “high stochasticity conditioning” to showcase the good performance of GSEE.

(3) The paper is well-written and accessible

**Weaknesses:**

(1) How would this uncertainty estimation metrics GSEE enhance the trustworthiness of generated outputs? For instance, could it be effective in detecting hallucinations and how does it perform? Additionally, what is the performance of using the GSEE to predict the accuracy of the output? Could you please show some examples or use experimental results to support the claim that the GSEE could be beneficial in hallucination detection and improving the prediction accuracy?

(2) What are the advantages of GSEE compared with other metrics like semantic entropy [1], which also measure the uncertainty from the semantic perspective.

(3) Is the GSEE metric sensitive to the number of generated responses M and temperature? It seems that as more responses are generated, the diversity among them would increase, potentially affecting the stability of the GSEE.

(4) The novelty of the definition of the metric is limited, as it relies on the existing metric to quantify the semantic diversity.

[1] Kuhn, L., Gal, Y., & Farquhar, S. (2023). Semantic uncertainty: Linguistic invariances for uncertainty estimation in natural language generation. arXiv preprint arXiv:2302.09664.

**Questions:**

Please see the weaknesses above.

---

> ### Author Response · Authors · 2024-11-27
> **Authors reply to Reviewer 4pJg**
>
> We greatly appreciate the reviewer’s feedback. In reply to the stated weaknesses:
> 1. Ideally, the GSEE-based UQ could be leveraged to help detect and mitigate hallucinations, as you mention, or to provide better calibrated model confidence estimates, or to augment RAG and other prompting techniques, or to provide a guidance in continual learning paradigms to address knowledge gaps in the LLM, etc. Our current experiments reflect strong correlations with GSEE-based UQ and generative “correctness” (measured with respect to Rouge-L and Perplexity metrics), in addition, we devise a less metric sensitive experimental setting using in-distribution vs. OOD prompts to further validate the effectiveness of GSEE as a bellweather for correct predictions. We believe that  GSEE can be leveraged additionally to optimize prompt engineering (e.g., design prompts to minimize uncertainty) to further improve model prediction, but we leave this exploration to future work.
> 2. Thank you for the question. In fact, we compare lengthy-normalized entropy as introduced in Kuhn et al., in our experimental baseline comparisons.
> 3. Yes, GSEE can be sensitive to the number of generated responses and temperature; we explore this sensitivity in several of our ablation experiments, please see: Table 6.
> 4. Thank you for the comment. While matrix entropy has been introduced in previous work to assess dataset complexity, to the best of our knowledge, it has never been previously applied for LLM and related model UQ purposes.

---

### Official Review · Reviewer_WrkJ · 2024-11-04

**Soundness:** 3
**Presentation:** 2
**Contribution:** 3
**Rating:** 5
**Confidence:** 3

**Summary:**

The paper introduces Generative-Semantic Entropy Estimation (GSEE), a model-agnostic algorithm for uncertainty quantification (UQ) in large language models (LLMs) and multi-modal models. GSEE estimates predictive uncertainty by analyzing the spectral entropy of covariance matrices derived from the latent embeddings of multiple generated responses. This approach aims to detect hallucinations and low-confidence predictions by measuring semantic diversity.

**Strengths:**

1. The method introduces a new approach to UQ by leveraging spectral entropy of covariance matrices derived from latent embeddings of generated responses.
2. GSEE’s model-agnostic design makes it applicable across various foundation models without the need for additional natural language inference (NLI) steps, enhancing its usability and adaptability to different contexts and models.
3. The paper provides a well-rounded evaluation of GSEE across multiple datasets

**Weaknesses:**

1. The method relies on latent embeddings extracted from penultimate layers, yet the impact of this choice is not examined. A detailed ablation study on embedding layer selection could clarify whether GSEE’s performance is sensitive to embedding depth, and if alternative embeddings could improve its accuracy.
2. Despite claiming efficiency, GSEE’s use of multiple generations and covariance calculations could be computationally intensive for larger datasets or more frequent real-time applications. The paper lacks a clear analysis of the computational costs and memory overhead of GSEE, which could hinder its scalability.
3. The presentation can be unclear and should be improved.

**Questions:**

Can the authors provide more details on the computational costs associated with GSEE, especially in comparison to simpler UQ methods? Insights into the time and memory complexity of GSEE could help clarify its practicality for large-scale deployment.

---

> ### Author Response · Authors · 2024-11-27
> **Authors comment to  Reviewer WrkJ**
>
> Thank you kindly to the reviewer. In reply to the stated weaknesses:
> 1. We agree with the reviewer that the uncertainty quantification can be critically dependent upon the manner and locus of latent embedding extraction. For our experiments, we chose to follow research “best practices” (as referenced) by averaging token likelihoods based on the penultimate layer model embeddings. In order to achieve an apples-to-apples comparison with other baseline UQ methods, we used this same embedding extraction method across all relevant methods. While we agree that an ablation study with regard to embedding extraction methods can provide some additional insights — particularly for determining a best extraction point for a given LLM/LMM — we nevertheless felt that this exploration was orthogonal to directly validating the effectiveness of GSEE vis a vis other LLM UQ methods, as the choice of embedding extraction locus is more suited as an application decision.
> 2. Indeed, this is a fine point. While creating multiple generations (M) for each model input can be parallelized through a single forward pass, the memory cost can be significant, particularly as GSEE is a white-box UQ method. This memory cost scales effectively at O(M); however, in practice, we found that using a reasonably small generation size, e.g., M=5, 10, produced favorable UQ results, which can be accommodated in many cases by a single GPU for many SoTA LLM/LMM models. With regard to covariance calculations, the reviewer’s concern is also well-founded, as SVD complexity is generally O(M^3); nonetheless, in our case, as the generation size is reasonably small and the covariance matrix calculations are with respect to MxM matrices, so we found this calculation to be close to negligible in comparison to model inference cost.
> 3. Please alert us to how we can make the presentation clearer, thank you.

---

### Official Review · Reviewer_iyRL · 2024-11-04

**Soundness:** 2
**Presentation:** 3
**Contribution:** 3
**Rating:** 6
**Confidence:** 3

**Summary:**

The paper proposes a method to evaluate the uncertainty of a language model, namely GSEE.
Specifically, the method generates a few outputs from an input prompt, then the method calculates the length-normalized spectral entropy of those outputs' semantic as the uncertainty.
The paper further evaluates the proposed methods under varied settings and compares the proposed method with baselines to show the superior performance of the proposed method.

**Strengths:**

(1) The paper gives a straightforward method to calculate the uncertainty of a language model and gives good illustration and intuition in Fig. 1.

(2) The paper empirically shows the proposed method works well under varied settings.

**Weaknesses:**

(1) It's a bit weird for me since the uncertainty is calculated based on a set of outputs. I'm not familiar with the literature but I have seen some papers working on uncertainty estimation which estimates the uncertainty of a particular output/generation. I feel it makes less sense to estimate the uncertainty of a set of generations since we usually just care about the uncertainty of a particular generation. (I acknowledge that it makes sense to estimate the uncertainty of a set of generations if we care about the uncertainty in a set of generations.)

**Questions:**

As weakness.

---

> ### Author Response · Authors · 2024-11-27
> **Authors reply to Reviewer iyRL**
>
> We thank the reviewer for their valuable feedback. In reply to the stated weakness: more formally, we wish to estimate the uncertainty of the posterior distribution p(y|x,t), where x denotes the input data (e.g., image+prompt), t is the model temperature, and y is the generated output. Effectively, each generated output is a sample drawn from p(y|x,t), and thus the dispersion of this sample (in our algorithm measured with matrix entropy) corresponds with predictive uncertainty; in general, this framework for modeling predictive uncertainty is well-established in literature, including Bayesian Neural Networks, see:
>
> Andrew Gordon Wilson and Pavel Izmailov. 2020. Bayesian deep learning and a probabilistic perspective of generalization. In Proceedings of the 34th International Conference on Neural Information Processing Systems (NIPS '20). Curran Associates Inc., Red Hook, NY, USA, Article 394, 4697–4708.

---

### Meta-Review · Area_Chair_XKEF · 2024-12-22

**Metareview:**

While the high-level idea seems interesting, the paper lacks clarity regarding its core claims, and key experimental details are missing or insufficiently justified, undermining confidence in the reported results. The rationale behind certain methodological choices remains unclear, and stronger evaluations are needed to show effectiveness. Overall, the paper requires more rigorous analysis and better exposition before it can be considered ready for publication. The reviewers have provided detailed feedback, which could help the authors improve their manuscript for future submissions.

**Additional Comments On Reviewer Discussion:**

See the metareview.

---

### Decision · Program_Chairs · 2025-01-22

Reject